# First Glimpse on Spring Starflower Domestication

**DOI:** 10.3390/genes13020243

**Published:** 2022-01-27

**Authors:** Agostina B. Sassone, Frank R. Blattner, Liliana M. Giussani, Diego H. Hojsgaard

**Affiliations:** 1Darwinion Institute of Botany (IBODA, CONICET-ANCEFN), Labardén 200, San Isidro B1642HYD, Buenos Aires, Argentina; lgiussani@darwin.edu.ar; 2Leibniz Institute of Plant Genetics and Crop Plant Research (IPK), Corrensstraße 3, 06466 Gatersleben, Germany; blattner@ipk-gatersleben.de; 3Albrecht-von-Haller Institute for Plant Sciences, University of Göttingen, Untere Karspüle 2, 03773 Göttingen, Germany

**Keywords:** Allioideae, Amaryllidaceae, cultivation, GBS, *Ipheion uniflorum*, SNPs

## Abstract

The cultivation and domestication of plants are human-driven processes that change the biology and attributes of a plant. *Ipheion uniflorum* is a bulbous geophyte known as Spring Starflower whose cultivation dates back to the first half of the 19th century. At least seven cultivars have been developed from natural stands. However, comparative analyses of wild and cultivated materials are largely missing. In the present study, we provide a morphological evaluation and analyses of the cytological and genetic variability of *I. uniflorum* that reveal significant levels of differentiation and evidence of artificial selection in the Spring Starflower. Distinctive phenotypic characters in cultivated materials that are rarely found or lacking in wild plants and natural populations, such as pink or violet flowers, together with its reduced heterozygosity and starting genetic differentiation support the view of early mechanisms of domestication acting upon Spring Starflower plants. The probable geographic origin of the cultivated forms is discussed together with perspectives for plant breeding.

## 1. Introduction

Domestication is the result of an evolutionary process in which wild species are exposed to new selective environments related to human cultivation and use, partly resulting in the increased fitness of the domesticated species [1]. Thus, domestication is a co-evolutionary, mutualistic process, in which domesticated species are transformed, adapted to human control, and propagated for human benefit, while humans start to become dependent on their domesticates. As domesticated plants became mostly cultivated and established from a relatively small number of founders, there are strong demographic effects (e.g., reduced effective population size, loss of diversity), and effects directed by the conscious or unconscious selection of traits that leads to genetic changes and dramatic impacts on genomic architecture [2]. Such changes drive the acquisition of unique adaptations relative to wild progenitors.

There are approximately 2000 semi- and fully domesticated plant species from 140 taxonomic families [3,4]. Most are relatively young, in some cases existing for just a few centuries. Most domesticated plant species grown for aesthetic enjoyment entered cultivation after 1500 AC [5]. Unlike domesticated crops, such as rice or grapes, in which domestication has been a protracted process driven in large part by unconscious selection of traits, such as seed size, seed retention (non-shattering), loss of seed dormancy or synchronous germination [6,7], and long and gradual periods of effective population size reduction, ornamental domestication was initiated more recently and is largely driven by the conscious selection of aesthetic traits, such as leaf or flower color, size or shape, through human choice and design [1,8]. In the case of *Ipheion uniflorum* (Graham) Raf. (Amaryllidaceae, Allioideae), ornamental forms and cultivars have been developed that resulted in the acquisition of new phenotypic traits and are widely known as Spring Starflower.

*Ipheion uniflorum* is a South American bulbous geophyte endemic to the Pampas in Argentina (Buenos Aires), Uruguay and South Brazil [9,10], and one of the most well-known and widely grown of South American bulbous ornamentals [11]. Naturally, it grows in open grasslands and rocky fields, in sun to partial shade. During late winter and early spring, the bulbs produce grass-like foliage, up to 25 cm tall, and multiple flower scapes with single white, blueish or light purple star-shaped flowers [12]. The plants multiply sexually by cross-pollinated seeds, and asexually by bulbils, become dormant after blossoming, and can tolerate drought during the dormant season (summer and autumn). Because of its floral display, attractive flower, garlic-scented foliage, and the fact that is one of the first species to flower in spring, *I. uniflorum* has attracted the attention of the humans and has been cultivated since at least 1832 when the naturalist J. Tweedie sent specimens from Buenos Aires, Argentina to the U.K. This material was described under *Milla uniflora* Graham as bearing “segments with six dark lines, which are purplish green behind, lilac in front” [10]. Currently, this species is cultivated in Japan, New Zealand and countries in North and South America, Europe, Africa, and Australia [9,13,14,15,16].

The famous botanist Stearn [13] provided a description of the popular forms (informally named as varieties in subsequent publications) of Spring Starflower used by different gardeners and researchers to generate, define or characterize the currently known cultivars e.g., [10,15]. This includes at least seven cultivars discerned mainly by their floral characteristics, particularly by their color. *I. uniflorum* ‘Alberto Castillo’ has a robust, 6 cm large white flower [11], and according to Howard [12], it was a selection of *I. uniflorum* f. *album* (Weathers) Stearn grown from seeds despite being listed as a clone. *I. uniflorum* ‘Froyle Mill’ is a darker purple/magenta-flowered form grown from seeds of *I. uniflorum* f. *purpureum* [17]. Another cultivar grown from seeds of *I. uniflorum* f. *violaceum* (Voss) Stearn is *I. uniflorum* ‘Wisley Blue’ with pale violet flowers. *I. uniflorum* ‘Rolf Fiedler’ can have from pale-sky-blue-to-deep-periwinkle blue, sweetly scented 3–4 cm wide star-shaped flowers, and can grow from 10 cm up to 40 cm. *I. uniflorum* ‘Charlotte Bishop’ is distinguished by its light pink flowers, while the *I. uniflorum* ‘Jessie’ appears to be a selected clone of *I. uniflorum* ‘Rolf Fiedler’ with more intense blue flowers and darker midribs (https://www.pacificbulbsociety.org/pbswiki/index.php/Ipheion, accessed on 28 December 2021). In the last decade, a new cultivar with pink-to-violet flowers was released, marketed as *I. uniflorum* ‘Tessa’.

While in nature *I. uniflorum* plants exhibit a wide range of morphological differences, such as flower size, robustness of plants, presence of bulbils or flower color, cultivated materials show less morphological variation [11,12,18,19] and a few particularly interesting traits that are not found in natural populations, which reflect the initial steps of cultivation and the action of a (pre-)domestication process [20]. For instance, except for the dark blue color of the flowers described for the cultivar *I. uniflorum* ‘Rolf Fiedler’ and *I. uniflorum* ‘Jessie’, all the other plant features have also been observed as part of the stock of variation within natural populations, in some cases in single individuals. In addition, the flowers of the cultivar *I. uniflorum* ‘Charlotte Bishop’ and *I. uniflorum* ‘Tessa’ have incorporated pinkish and rose colors, hardly ever found in nature (Sassone and Giussani, pers. obs.; [11,12]). As much of the genetics underlying this variation has not yet been unraveled, a first exploratory analysis will undoubtedly be useful and much needed for more in-depth and comprehensive studies.

In this paper, we aim to provide a first insight into the initial steps of Spring Starflower domestication and possible underlying genetic-evolutionary mechanisms. To this end, we use a bottom-up approach looking for genomic signatures of selection from extant genetic and morphological variations. Thus, we characterize patterns of variation in genome sequence diversity, structure, and organization between wild and cultivated *I. uniflorum* plants, and give a first attempt to identify ancestry and causative genetic changes likely to be associated with genetic and phenotypic responses to artificial selection. The data collected provide useful information for the management of *I. uniflorum* genetic resources for ornamental improvement.

## 2. Materials and Methods

### 2.1. Plant Materials

A total of 56 wild individuals of *Ipheion uniflorum* were used in the current study, including all individuals available from Sassone et al. [19], plus 5 individuals of different cultivated origin. Following the geographic differentiation within *I. uniflorum* previously detected [19], we structured our sampling of individuals into three geographic regions (South, Centre and North; see Appendix A) and two conditions (natural and cultivated). In addition, to contrast our sampling against the natural distribution of *I. uniflorum* and identify possible sampling bias, we downloaded the available information in GBIF [under the name: *I. uniflorum* (Lindl.) Raf] and kept only specimens bearing geographical coordinates (Pampean region, country codes: AR, BR and UY) [21]. The map was constructed in the R environment with the packages maptools v. 0.9-8 [22] and ggplot2 v. 3.3.3 [23].

Among the cultivated materials, two individuals were sampled in a private garden of Los Andes, Chile (SSF1 and SSF2), which were established from bulbs imported from England. One of these plants was identified as *I. uniflorum* ‘Rolf Fiedler’ (SSF2). A third individual was collected in a garden in San Isidro, Buenos Aires, Argentina (SSF3). The last two (SSF4 and SSF5) specimens were cultivated at the Botanical Garden and Botanical Museum in Berlin, Germany. All cultivated samples were compared against all 56 wild individuals from 10 collection sites (corresponding to 13 natural populations) distributed across the natural range of the species (Appendix A). Both the cultivated and wild individuals were kept alive in a glasshouse at the Darwinion Institute of Botany (Buenos Aires, Argentina) and the Leibniz Institute of Plant Genetics and Crop Plant Research (IPK, Gatersleben, Germany). Voucher specimens from each collection site were stored at SI.

### 2.2. Morphological Assessment

Morphological exploration focused on identifying traits likely selected in cultivated plants and of potential interest for horticulturalists. Flower color, shape of the tepal apices and the size of the flower were studied in individuals from the natural populations. We also collected extra information about the analyzed traits from an exhaustive review of the literature [9,11,12,16] and commercial websites marketing the cultivars. In addition, flower color variability and tepal apices were also evaluated in ca. 1400 iNaturalist records (https://www.inaturalist.org/, accessed on 27 December 2021), 60 records from wild materials and ca. 1340 from cultivated materials (>1000 from the U.S.A., ca. 300 from Europe, and the rest from Australia, Japan, New Zealand and South Africa). The collected data are summarized in Appendix A, including the codification of the morphological characters.

### 2.3. Karyological Analyses and Ploidy Estimations

Fresh root tips were collected and treated for 6 hr at room temperature with a solution of 0.05% colchicine, fixed for 12–48 hr in ethanol/acetic acid (3:1, *v*/*v*) and stored in 70% ethanol at 4 °C. Root tips were then hydrolysed for 10 min at 60 °C in a solution of 1 m HCl, and stained for 20 min or more using Schiff’s reagents (Feulgen reaction; Sigma-Aldrich, St. Louis, MO, USA). Root meristems were macerated in a drop of 2% acetic orcein and analyzed using a Leica DM5500B microscope (Leica Microsystems GmbH, Wetzlar, Germany). Chromosome measurements were performed using three to six selected mitotic metaphases with the Leica Application Suite software (LAS version 4.1.0, Leica Microsystems AG, Heerbrugg, Switzerland). Chromosomes were classified as metacentric (r = 1.00–1.49), submetacentric (r = 1.50–2.99), acrocentric (r ≥ 3.00) or telocentric (r = ∞), where r = length of the long arm/length of the short arm.

Karyological differentiation among individuals was inferred by the mean length of chromosome complements and karyotype symmetries. For possible interchromosomal asymmetries, the A2 index [24] was used. For intrachromosomal asymmetries, the total form percentage (TF%; [25]), the karyotype asymmetry index percentage (Ask%; [26]), the symmetric index (Syi; [27]) and the intrachromosomal asymmetry index (A1; [24]) were estimated. In addition, Stebbins’ categories [28] were added for the characterization of intrachromosomal asymmetries. For each asymmetry index, the variation between mean values of natural and cultivated individuals was contrasted statistically by ANOVA followed by Tukey’s test.

Ploidy estimations were based on cytological analysis (above) and by measures of genome-wide heterozygosity [19]. Briefly, the R package “gbs2ploidy” [29] was used to infer ploidy of individuals based on the allelic ratios of heterozygous single nucleotide polymorphisms (SNPs) during variant calling. Cases of inferred ploidy were corroborated using samples with previous cytological data and known ploidy.

### 2.4. Genotyping by Sequencing (GBS) Analyses and Population Genomics of Natural and Cultivated Materials

DNA extraction, library preparation and sequencing were carried out as described in Sassone et al. [19]. GBS loci were assembled using the ipyrad version 0.9.77 pipeline [30]. The different parameters were defined after exploring alternative settings following the recommendations of Eaton et al. [31,32]. Sequence reads for the GBS Illumina runs are stored in the European Nucleotide Archive (ENA, http://www.ebi.ac.uk/ena/data/view/PRJEB43919, accessed on 29 December 2021).

A discriminant analysis of principal components (DAPC, [33]) was carried out using the detected number of clusters of natural populations (three groups [19]). The DAPC first performs a principal component analysis (PCA) to summarize genotypic variation among individuals, and then a discriminant analysis to categorize the PCA results, thus maximizing the variation among a predefined set of groups and minimizing variation within them. The DAPC was also used to identify the genomic similarity of the five cultivated specimens to each other and to the samples collected in nature. The analyses were performed using the R package ‘adegenet’ [34] and confirmed through a cross-validation test with 1000 permutations. The first three principal components (48.46% of variance conserved) of the PCA and three discriminant eigenvalues were retained. These values were confirmed by cross-validation analysis. The genetic diversity among accessions was also assessed with the phylogenetic network using the neighbor-net method [35]. The Bayesian clustering method STRUCTURE version 2.2.4 [36] was used to determine the number of distinct genetic clusters (K) and with a burn-in period of 500,000 followed by 2,000,000 repetitions, as implemented in the ipyrad API. Ten replicate analyses were performed with values of K = 1–10. Population genetic summary statistics (*H*_O_, *H*_E_, *H*_T_, *F*_ST_, and *F*_IS_) were calculated to describe and compare overall and population-specific genetic diversity using the R package ‘hierfstat’ [37], and file conversions were facilitated with the R package ‘dartR’ [38]. We also used the R package ‘hierfstat’ to calculate the pairwise genetic distance (*F*_ST_) for each of the subpopulations, according to Weir and Cockerham [39].

## 3. Results

### 3.1. Morphological Characters within Ipheion uniflorum

When assessing qualitative characters among natural versus cultivated materials including the revised literature, an interesting bias in the distribution of characters was found. While natural *Ipheion uniflorum* plants generally have white flowers (in 86% of records, *n* = 43) and lilac or blueish flowers (these colors appear in some individuals within the populations, Appendix A), sky-blue-like flowers with rounded tepals have only been reported in two populations from Uruguay (Punta del Este and Piriapolis [11]) and they resemble the deep periwinkle blue flowers that are a distinctive trait of the *I. uniflorum* ‘Rolf Fiedler’ (Figure 1). This color is only found in Spring Starflowers. Likewise, pink and pinkish colors exhibited in the *I. uniflorum* ‘Jessie’ and ‘Tessa’ as well as the violet colors of *I. uniflorum* ‘Tessa’ were not recorded in our dataset (Appendix A). Among the studied materials, variation in tepal apices morphology (acute or apiculate, obtuse or round apices) is also found across natural populations, the obtuse tepal apices being the most widespread (Figure 1).

When assessing quantitative characters, the size of the flower exhibited a broad variation within the species from 2 to 3.5 cm length with a tepal fusion from 0.9 to 2 cm (Appendix A). In addition, our assessment also indicates that the variation in morphological characters found among wild individuals is not significantly different from that observed among cultivated individuals and show no extreme values except for a few plant size measurements (Appendix A). Data from different studies and growers (Appendix A) indicate that cultivated plants are taller, often reaching up to 30 or 40 cm height, whereas wild plants sizes range from 6 to 20 cm height.

### 3.2. Cytological Variability within I. uniflorum

Cytological information of *I. uniflorum* was gathered from this study and previous analysis ([19] and references therein) for six natural and five cultivated specimens (Figure 2; Table 1). Overall comparative analyses of ploidy, genome size and karyotype formula were highly stable among individuals, except for the finding of a polyploid individual among the cultivated materials (sample AS183; Table 1). Detailed karyological data were extracted for seven individuals from the South and Centre geographic areas, while the very few individuals available from the northern region and from cultivated materials did not provide enough good quality metaphases. The different chromosomal parameters support the cytological data of Table 1 and add sources of karyological variation between individuals from the South and Central regions.

The quantification of the heterogeneity in chromosome size through intrachromosomal asymmetry indexes showed little variation between individuals and regions, all cases displaying A1 = 0.99, falling in the 1C Stebbins category, and non-significant differences for TF%, Ask% and Syi indexes (Table 2). The interchromosomal asymmetry index A2 also displayed non-significant differences with values close to zero for individuals in the southern (A2 = 0.077) as in the central (A2 = 0.093) areas, reflecting conservation among chromosome size in the karyotype. Yet, variation in some chromosomal parameters from individuals between regions was observed. The largest/smallest chromosome ratio (R) were more variable among individuals from the South (R = 4.0; range 2.3–4.9) in comparison to individuals from the central region (R = 3.4; range 3.1–3.6). Likewise, a wider variation was found in the mean chromosome ratio (long/short arms; r) between individuals from the southern (4.2 ± 0.91, min 3.1, max 4.7) compared to those from the central (4.0 ± 0.27, min 3.6 max 4.2) regions (Table 2), as well as the number of chromosome pairs with arm ratio > 2 (r > 2), ranging 2–4 and 2–3 between individuals from the South and the Centre, respectively.

### 3.3. Genomic Diversity within I. uniflorum

In total, 20,641 loci were retained after filtering, and the concatenated GBS matrix was 1,864,491 bp length. Bayesian clustering analysis in STRUCTURE based on 11,355 SNPs (2302 unlinked) suggested that the best grouping number following ΔK was K = 3 (Appendix A), and this is mostly in concordance with the 3 geographical areas identified in a previous study [19] (Appendix A).

The DAPC was performed with four a priori groupings (three corresponding to the geographically differentiated natural populations plus one more for cultivated specimens (Figure 1).

The results show that the genetic similarity among wild individuals from natural populations and cultivated specimens is congruent among methods (Figure 3 and Appendix A). The cultivated materials SSF1and SSF2 show a genomic composition matching specimens from populations in the Centre of *I. uniflorum* distribution (LP, AS271, AS284; TA541, AS202) and reveals high levels of admixture with Northern populations (UR; AS545) (Figure 3a and Appendix A). Regarding SSF3, this genotype shows a genomic composition and level of admixture comparable to those found in Southern *I. uniflorum* populations (BB, AS135, AS138; MBu, AS260) (Figure 3a and Appendix A). The other two genotypes, the SSF4 and SSF5, show a distinct pattern with slight differences between methods. While in the STRUCTURE analysis these cultivated materials are grouped with wild individuals from Southern/Centre populations (Appendix A), the DAPC shows them as clearly differentiated from wild individuals in all populations (Figure 3a).

The neighbour-net analysis depicts high genetic diversity and a phylogenetic network concordant with the above results (Figure 3b). The analysis also shows the genotype SSF1 segregated from individuals from the Centre and more connected to individuals from the South (SCa; AS236, AS298, AS297) (Figure 3b). The genotype SSF3 is connected to Southern populations, but also it appears close to one individual from the Centre (SA; AS188) (Figure 3b). Despite SSF4 and SSF5 remain segregated, they are far away connected to an individual from a population of the North of *I. uniflorum* distribution (UR; AS545) (Figure 3b).

The summary statistics resulted in the values of *H*_O_ = 0.071, *H*_E_ = 0.196, *H*_T_ = 0.284, *F*_ST_ = 0.213 and *F*_IS_ = 0.636 for the species, including cultivated specimens (*p* < 0.001). For the Spring Starflower subgroup alone, the statistics give values of *H*_O_ = 0.051, *H*_E_ = 0.090, *F*_IS_ = 0.340. The estimates of pairwise fixation indices between regional and cultivated groups revealed the strongest differentiation for cultivated specimens with southern populations (*F*_ST_ = 0.485; Table 3). The comparison with the central populations showed an intermediate level of differentiation (*F*_ST_ = 0.249; Table 3), while with northern populations, the observed level of genetic differentiation was the lowest (*F*_ST_ = 0.139; Table 3).

## 4. Discussion

Domestication is a process involving the accumulation of trait differences over time, and, therefore, domestication mechanisms are most evident and widely known among the earliest and intensively cultivated species [2]. Domestication has dramatic impacts on genomic architecture reflecting historical demographic transitions (e.g., small number of founders and decreased effective population sizes) and changes in gene flow (e.g., directed selection) [2]. In more recently or less intensively cultivated plants, such as ornamentals, trait differences to wild relatives are expectedly less evident, and the genomic changes driven by incipient domestication processes are less known. In this paper, we present an initial analysis aimed at understanding the effects of human cultivation and artificial selection on *Ipheion uniflorum* that suggest the action of early domestication mechanisms in the Spring Starflower.

### 4.1. Natural Variation within Ipheion uniflorum

*Ipheion uniflorum* is the most relevant species within the genus *Ipheion*. From an evolutionary viewpoint, the species shows the broadest distribution, largest genetic and morphological diversity, and has played a pivotal role in the origin of its sister species [19]. From an economic viewpoint, the species is a resource of the different cultivated types of Spring Starflower.

The species grows in the Pampas of South America, a region subjected to intensive agricultural activities. Pampa’s grasslands have only been preserved in areas with unfavorable edaphic or climatic conditions for agriculture, such as hills (Ventania and Tandilia systems), floodplains (e.g., the Salado basin) or at rural roadsides [40].

The observed within (and among) population morphological variability (e.g., Figure 1, Appendix A) is more relevant in traits such as flower color and size, plant robustness and the presence of bulbils [11,12], and has support in extant cytological and molecular diversity. Even though the species is karyologically and genetically differentiated from the sister species *I. tweedieanum* (Griseb.) Traub and *I. recurvifolium* (C.H.Wright) Traub, at the intraspecific level, no chromosomal structural changes between the studied *I. uniflorum* samples and minor differences in cytological and karyological parameters were found, particularly among individuals from the South and Centre of the distribution (Figure 2; Table 1 and Table 2). In the genetic variation analysis, the use of Evanno’s ΔK method and best-fit population models at K = 2 and 3 revealed three genetically differentiated regional groups that follow a latitudinal geographical distribution [19]. Individuals in populations from the South (BB, MBu, SCa), Centre (MP, LaPer, SA, AZ, TA), and North (UR, LP) of the species distribution show substantial differences in private loci and relatively low admixture among them [19]. Wild *I. uniflorum* individuals across populations showed values of heterozygosity (*H*_O_ = 0.084) lower than expected (*H*_E_ = 0.178), with overall levels of genetic diversity *H*_T_ = 0.267. The fixation indices and inbreeding coefficients showed values of *F*_ST_ = 0.334 and *F*_IS_ = 0.527, respectively, with the strongest differentiation decreasing from south towards north [19]. Overall, the data support the view that the natural variability of *I. uniflorum* in morphological, karyological, and genetic attributes being geographically structured with a gradual latitudinal differentiation trend.

### 4.2. Evidence for Early Domestication Mechanisms in I. uniflorum

As mentioned above, the domestication of ornamental plants was initiated more recently than other intensively cultivated crops and is largely dependent on aesthetically driven human selection [1,8]. This implies that morphological changes in aesthetic traits by human choice and design and accompanying changes in the genomic architecture will show a variable incidence of the domestication syndrome depending on the cultivation and domestication history of each ornamental.

Ornamental forms and cultivars of *I. uniflorum* have a recent origin dating back to the first half of the 19th century. Despite the relatively recent cultivation start, *I. uniflorum*’s cultivars have acquired new phenotypic traits that differentiate to a certain extent the cultivated materials from the wild ones. The most striking one relates to flower color and the shape of tepal apices. The cultivar *I. uniflorum* ‘Rolf Fiedler’ displays a deep periwinkle blue flower color in combination with rounded tepals apices that only bear a resemblance to individuals from two Uruguayan populations (but see comments in the next section). Similarly, the distinctive pink flower color of the cultivar ‘Jessie’, or the pink-to-violet flower color of the recently released cultivar ‘Tessa’ have not been observed among natural populations.

While in nature, plant sizes range from 6–20 cm height, individuals in cultivation often reach up to 25–40 cm. Phenotypic variation for quantitative genetic traits, such as plant height, is continuously distributed in natural populations and often responds slowly to artificial selection [41,42]. In our study, the tendency to bigger plant sizes observed in Spring Starflowers might suggest a visible effect of artificial selection on genes influencing vegetative growth. Of course, an effect of nutrient availability in soils used for cultivation cannot be discarded, but we can point out that soils in the geographic range where *I. uniflorum* grows naturally are among the richest worldwide [43]. So, if there is an effect of nutrient availability, we might expect it to be low. This assumption could be easily tested in future studies by growing and evaluating different accessions under the same environmental conditions.

Regarding the genetic architecture of the Spring Starflower materials, our population structure analysis shows the cultivated materials are integrated among the three regional genetic clusters (Appendix A). Whereas the wild *I. uniflorum* populations show best-fit population models at K = 2 and 3, when adding the cultivated specimens, Evanno’s ΔK shifts the best-fit population model to K = 3, enhancing the differentiation between North and Centre populations (Appendix A).

This agrees with the DAPC and neighbor-net analyses that place three of the cultivated genotypes in-between individuals from the different regions by holding variable levels of admixture and segregates the other two (Figure 3a,b). For these two cultivated individuals (SSF4 and SSF5), the DAPC analysis identifies a new genomic composition of sequences that differs from the explored wild populations (Figure 3a,b). This indicates substantial genetic differences between these two cultivated genotypes and the natural genotypes studied in this paper. Whether SSF4 and SSF5 are derived from an otherwise extinct population/genotype no longer found today remains an open question.

Genetic diversity parameters indicate that, by including five cultivated specimens, the values of heterozygosity observed among natural populations (*H*_O_ = 0.084) exhibit a considerable decline (*H*_O_ = 0.071), but the expected heterozygosity (*H*_E_ = 0.178) and the overall levels of genetic diversity (*H*_T_ = 0.267) show a slight upsurge (*H*_E_ = 0.196, *H*_T_ = 0.284). Likewise, the relative fixation indices are reduced from *F*_ST_ = 0.334 [19] to *F*_ST_ = 0.213, and the inbreeding coefficient changes from *F*_IS_ = 0.527 [19] to *F*_IS_ = 0.636. On the one hand, the data suggest a slight reduction in the genetic diversity of the dataset after including the cultivated samples, likely due to inbreeding. On the other hand, the gain in overall genetic diversity and decay in fixation indices indicate some level of genetic differentiation, which agrees with the observed segregation of two Spring Starflower specimens from the natural *I. uniflorum* populations mentioned above.

The overall data on morphological traits and genome sequence diversity point to the occurrence of phenotypic responses to artificial selection in the cultivated genotypes and suggest the action of early domestication mechanisms as a probable side effect of human cultivation.

Among cultivated species, gradual declines in effective population sizes and genetic bottlenecks often occur during initial domestication, and a further reduction in genetic diversity is caused by artificial selection [1,2,44]. In our analysis, a substantial reduction in levels of heterozygosity in cultivated Spring Starflower relative to wild materials was found. While in domesticated crops it is possible to distinguish between a decrease in genetic diversity caused either by a genetic bottleneck (genetic drift results in genome-wide reduction in diversity) or by artificial selection (expected to reduce diversity in domestication-associated genomic regions) (e.g., [45,46]), in our case it was not possible to draw a precise conclusion. Considering that we are not aware of any active genetic improvement program using the Spring Starflower, the reduced genetic diversity observed in cultivated samples is likely caused by genetic drifts initiated through pre-domestication human management of wild stands. A wider sampling, including cultivated materials from diverse collections and markets, is needed to properly evaluate the extent of such genetic bottlenecks due to reduction in effective population sizes and the contribution of artificial selection to the observed decrease in heterozygosity.

### 4.3. A Glance at the Origin and Future of the Spring Starflower

Our genomic analysis provides little resolution on the original populations from which Spring Starflower cultivars used in the present study were first harvested. Yet, at least two cultivated individuals (SSF1 and SSF3) can be assigned to natural populations of a geographic group. One showing a genomic composition grouping with populations in the Centre and Northern regions (SSF1) and the other one with populations in the Southern region (SSF3). Most importantly, the proposed Uruguayan origin of the *I. uniflorum* ‘Rolf Fiedler’ [11] is corroborated, which here shows high levels of admixture to a single Northern population of *I. uniflorum* (UR; AS545).

A point that deserves consideration and is recurrent in color profiling for plant biology studies [47] is the lack of a systematic evaluation for the color of flowers found within *I. uniflorum* individuals. The occurrence of natural variation of floral color and the natural changes in flower pigmentation during flower development, maturation and climatic conditions makes it difficult to obtain standardized measurements of flower colors, which would help scientists to manage biodiversity stands. Therefore, without a numerical classification of flower colors obtained, for example, by a colorimeter (e.g., [48]), the current categorization of flower color in natural and cultivated plants remains subjected to human interpretation bias, especially when stating the occurrence of colors in the lilac-blueish range.

From a plant breeding perspective, the summary statistics of Spring Starflower specimens (*p* < 0.001) show low levels of heterozygosity (*H*_O_ = 0.051) compared to the expected value (*H*_E_ = 0.090), and relatively high inbreeding (*F*_IS_ = 0.340). However, these parameters might show bias due to the small sampling size. The pairwise fixation indices showed the strongest differentiation of cultivated specimens to populations in the South (*F*_ST_ = 0.485), and the lowest genetic differentiation with populations in the North (*F*_ST_ = 0.139). This pattern parallels the genetic differentiation trend observed among *I. uniflorum* natural populations [19]. Thus, wild populations of *I. uniflorum* might well work as reservoirs of morphological and genetic variability for the further development of Spring Starflower cultivars. The continued development of new cultivars (e.g., *I. uniflorum* ‘Tessa’) depict both the interest of horticulturalists and the appreciation of the public (four cultivars had won prizes of the Royal Horticultural Society, https://www.rhs.org.uk/plants/pdfs/agm-lists/agm-ornamentals.pdf, accessed on 27 December 2021). In an era of accelerated human-induced species losses, many natural populations are today threatened and may become extinct in the next few years if no conservation measures are taken. The adoption of measures to protect and conserve the remaining populations of *I. uniflorum* throughout the distribution of the species (perhaps considering geographic regions as separate genetic reservoirs) is of great importance for future breeding plans and the release of new cultivars.

## Figures and Tables

**Figure 1 genes-13-00243-f001:**
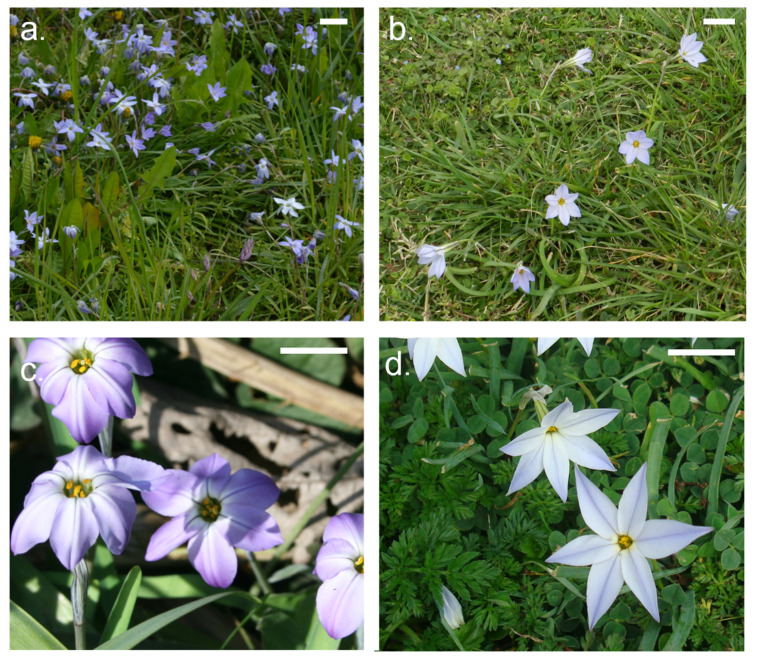
Plate depicting morphological variation of cultivated (**a**,**c**) and natural (**b**,**d**) materials. (**a**) *Ipheion uniflorum* Raf. at BGBM (Berlin). (**b**) Population of *I. uniflorum* from the Central region (Azul, Buenos Aires). (**c**) *I. uniflorum* ‘Rolf Fiedler’ from Ezeiza Botanical Garden (Argentina). (**d**) Plants of *I. uniflorum* from the Central region (Mar del Plata, Buenos Aires). Photo credits: L.M. Giussani and A.B. Sassone. The bar represents 2 cm (**a**,**b**) and 1 cm (**c**,**d**).

**Figure 2 genes-13-00243-f002:**
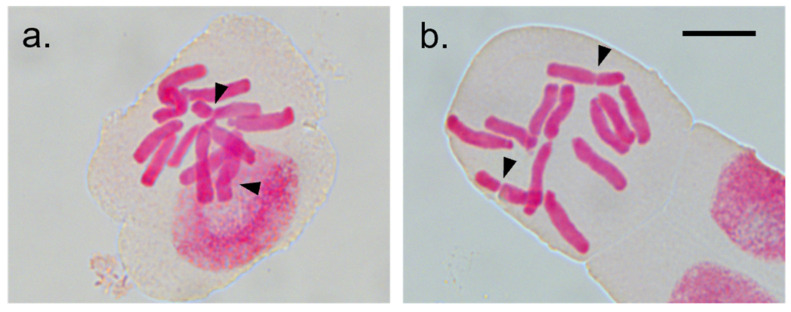
Mitotic metaphase cells of wild (**a**) and cultivated (**b**) *I. uniflorum* plants. (**a**) Sample AS98; (**b**) Sample AS247. Arrowheads identify centromeres of submetacentric chromosomes. The bar represents 10 μm.

**Figure 3 genes-13-00243-f003:**
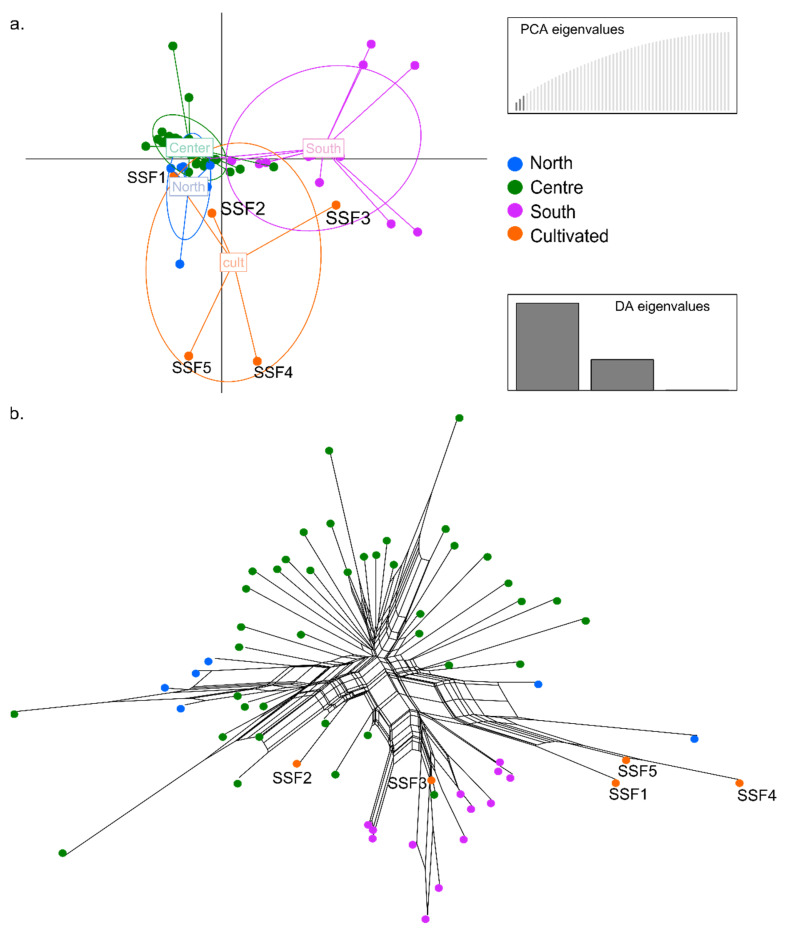
Genetic analyses of *I. uniflorum* materials. (**a**) Discriminant analysis of principal components (DAPC) for 61 accessions using only biallelic loci. The three PCs and discriminant eigenvalues retained describe the relationship between the clusters. Each axis represents a linear discriminant and the circles cluster individuals (represented by dots) by region or condition (natural or cultivated). (**b**) phylogenetic network analysis calculated using Splitstree4 using 17,782 unique SNPs. Each dot represents an individual and colors are as in (**a**).

**Table 1 genes-13-00243-t001:** Cytological analyses of natural and cultivated *Ipheion uniflorum* samples.

Population	Sample	Origin	Ploidy		Karyotype Formula	2C*x* (pg.)
			Cytology	gbs2ploidy		
513_uniAZ	AS203	Natural	2*n* = 12	2*x*	1SM + 5A	9.6
540_uniAZ	AS224	Natural	2*n* = 12	2*x*	1SM + 5A	---
553_uniTA	AS235	Natural	2*n* = 12	2*x*	1SM + 5A	9.65
555_uniTA	AS98	Natural	2*n* = 12	2*x*	1SM + 5A	10.05
S51_uniSCa	AS236	Natural	2*n* = 12	2*x*	1SM + 5A	---
Vil_uniBB	AS135	Natural	2*n* = 12	2*x*	1SM + 5A	---
SSF1	AS182	Cultivated	2*n* = 12	2*x*	1SM + 5A	9.6
SSF2	AS183	Cultivated	2*n* = 24	4*x*	1SM + 5A	18.8
SSF3	AS247	Cultivated	2*n* = 12	2*x*	1SM + 5A	9.75
SSF4	AS266	Cultivated	---	2*x*	---	---
SSF5	AS267	Cultivated	---	2*x*	---	---

**Table 2 genes-13-00243-t002:** Karyotyping parameters among diploid (2*n* = 2*x* = 12, 1SM + 5A) *I. uniflorum* individuals.

Region	TCL	c	c max	c min	i	TF%	Ask%	Syi	A1	A2	R	r
**Centre**	111.7 ± 9.9	9.33 ± 0.8	11.1	7.1	24.1	8.9	91.1	9.8	0.99	0.093	3.4	4.0
**South**	121 ± 9.6	10.1 ± 0.8	11.6	8	24.9	9.1	90.9	10	0.99	0.076	4.0	4.2

TCL = haploid complement length (µm); c = mean chromosome length (µm); c max, c min = maximum and minimum chromosome length (µm); i = mean centromeric index; TF% = ratio between the total sum of short arms (p) and the total length of a chromosome set ×100; Ask% = ratio between the total sum of long arms (q) and the total length of a chromosome set ×100; Syi = ratio between the mean length of the short arms (p) and the mean length of the long arms (q) ×100. A1, A2 = intrachromosomal and interchromosomal asymmetry indices, respectively; R = largest/smallest chromosome ratio; r = mean chromosome ratio (long/short arms).

**Table 3 genes-13-00243-t003:** Pairwise genetic differentiation values (*F*_ST_) for 61 accessions belonging to 4 clusters by region and condition (natural or cultivated).

	North	Center	South
**Center**	0.154 (0.147–0.147)	-	-
**South**	0.322 (0.314–0.314)	0.308 (0.302–0.314)	-
**Cultivated**	0.139 (0.128–0.149)	0.249 (0.242–0.248)	0.485 (0.477–0.493)

In brackets: lower and upper bound of confidence interval limits. In all cases, *p*-value < 0.001.

## Data Availability

Sample codes for ENA accessions (available from http://www.ebi.ac.uk/ena/data/view/PRJEB43919, accessed on 26 December 2021) among cultivated materials: AS182/SAMEA8419308, AS183/SAMEA8419309, AS247/SAMEA8419333, AS266/SAMEA8419338, AS267/SAMEA8419339. ipyrad parameters and output files of filtered GBS loci are available upon request.

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
