# Peer review of "First Glimpse on Spring Starflower Domestication"

_genes, 2022, doi:10.3390/genes13020243_

Round 1

Reviewer 1 Report

This manuscript represents some great work on understudied species, spring starflower. As a plant which has only recently come under cultivation, I. uniflorum may be, as the authors posit, useful for understanding the process of domestication. However, the definition of “domesticated” posited by the authors is somewhat confused. The authors refer to domestication as both a process and a state. While both understandings can be valuable, it confuses the narrative of the manuscript. If domestication is an ongoing, co-evolutionary process, it would be more correct to refer to the managed I. uniflorum plants as “under cultivation.” If, on the other hand, the authors wish to consider domestication to be a state, as opposed to the wild state, they do not provide sufficient evidence that I. uniflorum is indeed domesticated. While they do point out that cultivated I. uniflorum has decreased genetic diversity and differs in some physical phenotypes, this is not sufficient evidence for domestication. In sum, while this manuscript details the range of physical and genetic traits that occur within cultivated I. uniflorum, it is too far a stretch to refer to the cultivated plants as “domesticated.”

One of the specific claims that is used to differentiate wild from cultivated plants is the presence of colors not observed in the wild population, such as pink or purple flowers. The authors only report using 56 wild accessions. How these accessions were chosen was not explained, so the observed differences in phenotype could be simply the result of selection bias. Further, only five cultivated accessions were used as part of this study. How were these accessions chosen? The manuscript would benefit from a map detailing the origin locations of the tested accessions.

Other than the specific claims regarding the domestication of I. uniflorum and the lack of explanation regarding the chosen studied accessions, the manuscript is very well constructed, and I have only minor wording concerns as follows:

Throughout: Please maintain a single voice, either active or passive. Sometimes sentences alternate within a paragraph; this is disorienting to the reader.

59: there are dashes around “at least.” What do these indicate?

85: There are dashes around “to some extent.” What do these indicate?

105-109: Unclear how the accessions were chosen. Are the 56 wild accessions a subset of the 88 tested in the cited paper and the 5 cultivated not included in that paper? Or are all 61 tested accessions from the previous paper? Why was only this subset used instead of all 88 from the previous publication?

116: There is no explanation of the 13 natural populations. Is this a subset of the 20 from Sassone et al? If so, why are only these 13 populations used in this study?

184: What does “eventually lilac” mean? Do the colors of the petals change over time? Do older and younger plants show different colors?

Author Response

Reviewer 1

R1: …the definition of “domesticated” posited by the authors is somewhat confused. The authors refer to domestication as both a process and a state. While both understandings can be valuable, it confuses the narrative of the manuscript. If domestication is an ongoing, co-evolutionary process, it would be more correct to refer to the managed I. uniflorum plants as “under cultivation.” If, on the other hand, the authors wish to consider domestication to be a state, as opposed to the wild state, they do not provide sufficient evidence that I. uniflorum is indeed domesticated…In sum, while this manuscript details the range of physical and genetic traits that occur within cultivated I. uniflorum, it is too far a stretch to refer to the cultivated plants as “domesticated.”

Yes, domestication is both a process and a state in a similar way that two populations diverge and acquire the status of separate species and yet they keep evolving. In the case of I. uniflorum, the cultivated materials are undergoing the process of domestication, yet they are not yet “domesticated” as observed in crops but the differentiation between wild and cultivated plants that leads to a domesticated status is active. We do agree that I. uniflorum plants are better referred as “under cultivation” rather than “domesticated”, we made corrections accordingly and have used that terminology along the text.

R1: One of the specific claims that is used to differentiate wild from cultivated plants is the presence of colors not observed in the wild population, such as pink or purple flowers. The authors only report using 56 wild accessions. How these accessions were chosen was not explained, so the observed differences in phenotype could be simply the result of selection bias. Further, only five cultivated accessions were used as part of this study. How were these accessions chosen? The manuscript would benefit from a map detailing the origin locations of the tested accessions.

Thank you for this observation. We now included a map (Fig. S1) showing the natural distribution of the species and our studied populations. We have also rephrased the paragraph to clarify this point. In our knowledge of the genus and its natural occurrence (the whole geographic range is subjected to intense agriculture), we think our sampling is representative enough to have a good description of the phenotypic, cytological, and genomic variability. In addition, we would like to mention that extra information about the flower colour variability and tepal apices from 60 records occurring along the natural distribution of the species was extracted from herbarium specimens available in iNaturalist (as written in lines 137-141).

The cultivated materials included in the study were the ones available when the GBS analysis was carried out. As discussed in the text, the cultivated accessions were obtained from different sources and even though information about the locality is known for each case (lines 119-124), we had no a priori information about the population of origin (lines 443-444). The link of cultivated genotypes to specific populations or geographic areas is part of the results from this study.

Minor comments:

R1: Throughout: Please maintain a single voice, either active or passive. Sometimes sentences alternate within a paragraph; this is disorienting to the reader.

Apologies. We checked the manuscript and corrected the text.

R1: 59: there are dashes around “at least.” What do these indicate?

85: There are dashes around “to some extent.” What do these indicate?

We deleted the dashes in the first case and deleted the expression in the second case.

R1: 105-109: Unclear how the accessions were chosen. Are the 56 wild accessions a subset of the 88 tested in the cited paper and the 5 cultivated not included in that paper? Or are all 61 tested accessions from the previous paper? Why was only this subset used instead of all 88 from the previous publication?

The complete analysis is new. We used 56 wild accessions (also reported in a previous study) plus 5 cultivated accessions (not included before). These are all I. uniflorum samples available. The rest of the samples of wild plants (32) included in the previous study belong to other Ipheion species. The text was rephrased.

R1: 116: There is no explanation of the 13 natural populations. Is this a subset of the 20 from Sassone et al? If so, why are only these 13 populations used in this study?

13 populations are all populations available from I. uniflorum (it should be clearer now in the text and with the new Figure S1). The other populations from Sassone et al. belong to another Ipheion species.

R1: 184: What does “eventually lilac” mean? Do the colors of the petals change over time? Do older and younger plants show different colors?

We deleted “eventually”. We wanted to point out that lilac or blueish flowers are not that common as white flowers.

Reviewer 2 Report

This manuscript brings interesting information on a recently domesticated ornamental plant. However, it is difficult to fully review the paper since the supplemental data is not available at the mentioned site: www.mdpi.com/xxx/s1

In addition, I suggest to refer to comments written in the attached file.

Author Response

Reviewer 2

R2: This manuscript brings interesting information on a recently domesticated ornamental plant. However, it is difficult to fully review the paper since the supplemental data is not available at the mentioned site: www.mdpi.com/xxx/s1

We apologize for any possible misunderstanding. Our supplementary material was included as a pdf file, the link to the mentioned site was generated by default during the submission process. So, all the supplementary data is included in the pdf file labelled as “manuscript-supplementary”. 

In addition, we have now included a link to the sequence reads for the GBS Illumina runs, which are stored in the European Nucleotide Archive (ENA, http://www.ebi.ac.uk/ena/ data/view/PRJEB43919)

R1: In addition, I suggest to refer to comments written in the attached file.

Thank you for these observations. We have included all of them. The changes can be followed through the track changes option.

Reviewer 3 Report

In this study, Sassone et al. conducted the genetic-evolutionary and morphological variation between cultivated materials and wild species in Spring Starflower, which providing useful information for domesticated mechanisms and genetic changes associated with phenotypic responses to artificial selection. The results extended to understand the morphological and genetic variability for further development of Spring Starflower cultivars. Overall, the strategy of the study is sound and the manuscript is readable, but there are still some issues to be resolved at present version. The followings are comments for authors' consideration.

  1. If the content of the manuscript suit for the scope of the journal Genes? The study seem like belong to ornamental horticulture field.
  2. In results 3.1, the authors should provide detailed partial photoes of morphological characters including tepals, flowers characters, size and so on to demonstrate the symptom differences among the natural and cultivated materials.
  3. In results 3.2, photoes of the karyological analyses under microscope should be provided in this part.
  4. In results 3.3, for the genomic diversity within Ipheion uniflorum, the authors should provide solid evidences to exhibit different loci or SNPs in sequences of SSF1-SSF5. Moreover, a compared analysis about the genomic variation among Ipheion uniflorum should be provided to demonstrate phylogenetic tree.
  5. This study is a preliminary description in Ipheion uniflorum, the results in this manuscript seem weak duo to no molecular characterization in his study. The authors should provide more solid evidences to demonstrate the different phenotypes and genetic variability among domesticated and wild species.

Author Response

Reviewer 3

R3: If the content of the manuscript suit for the scope of the journal Genes? The study seem like belong to ornamental horticulture field.

Yes, indeed. Besides it is true that our manuscript could fit in horticultural journals, it is also true that it fits well in journals dealing with genetic and genomic analyses (our main analyses use Genotyping by Sequencing data, GBS).

R3: In results 3.1, the authors should provide detailed partial photoes of morphological characters including tepals, flowers characters, size and so on to demonstrate the symptom differences among the natural and cultivated materials.

We have added a scale bar in Figure 1 so that size differences between wild and cultivated flowers can be better visualized together with the colours. As explained in the text (see e.g., lines 200-218, lines 349-352) the other characters show a wide variation and no significant differences between wild and cultivated materials.

R3: In results 3.2, photoes of the karyological analyses under microscope should be provided in this part.

We added a new Figure (now Fig. 2 in the main text) showing images of mitotic metaphase cells of wild and cultivated I. uniflorum individuals used in this study.

R3: In results 3.3, for the genomic diversity within Ipheion uniflorum, the authors should provide solid evidences to exhibit different loci or SNPs in sequences of SSF1-SSF5. Moreover, a compared analysis about the genomic variation among Ipheion uniflorum should be provided to demonstrate phylogenetic tree.

We are not sure which other type of specific analyses might the reviewer being suggesting here. Our results are based on SNPs collected and identified from comparative genomic analysis. The data is resumed using statistical parameters as it is usually done in studies like this one (see section 3.3, e.g., lines 313-321). The GBS approach employed here generates sequence level information from the whole genome, then the data is curated to allow comparisons among samples and, the variations are resumed using genetic diversity parameters (HO, HE, HT, Fis, Fst) as well as Discriminant Analysis of Principal Components, Neighbour Net analysis and pairwise comparisons (e.g., Table 3). The phylogenetic network on Fig. 2b (now Fig. 3b) is also based on 17782 SNPs found among all samples (it is mentioned in the Figure’s legend). The genetic differences between wild and cultivated materials (based on SNPs) are those detailed in the results section.

R3: This study is a preliminary description in Ipheion uniflorum, the results in this manuscript seem weak duo to no molecular characterization in his study. The authors should provide more solid evidences to demonstrate the different phenotypes and genetic variability among domesticated and wild species.

As mentioned above, we do present molecular evidence based on genetic variability and genetic differentiation between wild and cultivated materials (see e.g., Fig. 2, Fig. S1; Table 3). Perhaps the reviewer wants to be more specific regarding the type of molecular characterization being requested. There is also a file with supplementary data (named “manuscript-supplementary”) which might contain the type of molecular information that is requested.

Regarding the phenotypic differences between wild and cultivated plants, they mostly relate to the occurrence of distinct colours in flowers. While classifying a certain colour might be subjective and requires specific approaches to standardize their analysis (see our comments in lines 451-460), we can be confident that new colours like pink or violet (found in cultivated plants) compared to the white and blue-light blue (of wild plants) are not biased interpretations and belong to the phenotypic differences observed between wild and cultivated materials.

Round 2

Reviewer 3 Report

The revision has been revised significantly, and my concerns were addressed well. I do not have any further request on the manuscript.